# Cell recruitment and the origins of Anterior-Posterior asymmetries in the *Drosophila* wing

**Rosalío Reyes[1,2], Rafael Rodriguez-Muñoz[1], Marcos Nahmad[1] ***

**1** Department of Physiology, Biophysics, and Neurosciences; Center for Research and Advanced Studies (Cinvestav), Mexico City, Mexico, **2** Interdisciplinary Polytechnic Unit of Biotechnology of the National Polytechnic Institute, Mexico City, Mexico

* marcos.nahmad@cinvestav.mx

**Data Availability Statement:** All data files are available without restrictions at the following repository: https://zenodo.org/records/13770487.

## Abstract

The mechanisms underlying the establishment of asymmetric structures during development remain elusive. The wing of *Drosophila* is asymmetric along the Anterior-Posterior (AP) axis, but the developmental origins of this asymmetry is unknown. Here, we investigate the contribution of cell recruitment, a process that drives cell fate differentiation in the *Drosophila* wing disc, to the asymmetric shape and pattern of the adult wing. Genetic impairment of cell recruitment in the wing disc results in a significant gain of AP symmetry, which results from a reduction of the region between longitudinal vein 5 and the wing margin (L5-M) in the adult wing. Morphometric analysis confirms that blocking of cell recruitment results in a more symmetric wing with respect to controls, suggesting a contribution of cell recruitment to the establishment of asymmetry in the adult wing. In order to verify if this phenotype is originated during the time in which cell recruitment occurs during larval development, we examined the expression of a reporter for the selector gene *vestigial (vg)* in the corresponding pro-vein regions of the wing disc, but our findings could not explain our findings in adult wings. However, the circularity of the Vg pattern significantly increases in recruitment-impaired wing discs, suggesting that cell recruitment may contribute to AP asymmetries in the adult wing shape by altering the roundness of the Vg pattern. We conclude that cell recruitment, a widespread mechanism that participates in growth and patterning of several developing systems, may contribute, at least partially, to the asymmetric shape of the *Drosophila* wing.

## Introduction

The developmental origins of complex asymmetric body plans are largely unclear. Since organ shape is closely related to function, there is considerable interest in understanding the developmental processes that guide the robust establishment of these asymmetries [1–5]. One of the most studied mechanisms in bilateral organisms are the signaling pathways responsible for left-right asymmetries during the early stages of development [6–8]. However, little is known about the establishment of asymmetric patterns and shapes using morphogenetic signals that diffuse symmetrically away from their source. The wing of *Drosophila melanogaster* is a

**Funding:** The author(s) received no specific funding for this work.

**Competing interests:** The authors have declared that no competing interests exist.

biological system where the mechanisms of growth and patterning have been extensively studied [9–12]. During the early stages of wing development, the tissue undergoes an Anterior-Posterior (AP) compartmentalization [13]. The posterior compartment becomes the source of Hedgehog (Hh) signaling, which triggers the expression of Decapentaplegic (Dpp) in a specific pattern abutting the AP border [14]. Dpp establishes a morphogen signaling axis that plays a pivotal role in guiding disc growth along the AP axis and determining the characteristic venation pattern of the adult *Drosophila* wing [14, 15]. The longitudinal veins 2 (L2) and 5 (L5) of the adult wing are directly determined by specific landmarks of Dpp patterning [16], but they are not symmetrically positioned with respect to the AP boundary [17]. Furthermore, the shape of the wing is remarkably robust, but clearly asymmetric. Thus, this system offers a useful model to investigate how asymmetries arise and are robustly established in developmental patterning. The first evidence on the origin of these asymmetries was hinted by the asymmetric pattern of phosphorylated Mad (pMad) which likely arise due to the asymmetric expression of the Dpp receptor, Thickveins (Tkv) [17, 18]. However, the compartment-specific role of Dpp in wing disc growth remains controversial. Zhang *et al.*, (2013) suggested that the Dpp gradient might control proliferation in the lateral region, but it is not essential for uniform proliferation along the AP axis [19]. In contrast, more recent work by Matsuda *et al.,* (2021) challenged this idea by showing that the anterior compartment does not require Dpp diffusion for growth, suggesting that the role of Dpp in disc growth is primarily in the central regions [20, 21]. Therefore, while Dpp drives positioning of veins L2 and L3, it is unclear how Dpp contributes to asymmetries in growth along the AP axis.

In contrast to the AP axis, signaling along the Dorsal-Ventral (DV) axis must be symmetrical with respect to the DV boundary to ensure that the dorsal and ventral layers of the adult wing are properly formed [22]. Gene expression patterns orchestrated by the AP and DV signaling axes are also instrumental for initiating planar polarization and inducing oriented proliferation, which collectively generate global forces influencing tissue morphology [23, 24]. Global forces are implicated in various morphological features, such as the formation of folds surrounding the pouch [25, 26] and the bending of the pouch at the beginning of the metamorphosis [26]. While the eversion process is guided by oriented cell division and cell intercalation [9, 10], the prevailing assumption is that the final wing shape primarily arises from the anchoring of wing edge cells to the cuticle, followed by retraction of the hinge [9, 10, 27]. However, the intricate relationship between larval developmental processes and the final adult wing shape remains unclear.

AP and DV signaling are integrated through the expression of the wing selector gene *vestigial* (*vg*), which directly depends on Dpp and Wingless (Wg) signals [28–30], through a process known cell recruitment, which eventually results in the induction of undifferentiated cells to express *vg*. The molecular mechanism that activates cell recruitment begins with the down-regulation of the protocadherin Dachshous (Ds) in Vg-expressing cells. This induces a polarization in the distribution of Ds and Fat (Ft), another protocadherin that interacts heterotypically with Ds, at the boundary of the Vg pattern. The Ft-Ds dimer drives the activation of the Hippo pathway in the newly-recruited cell, which in turn induces the translocation of Yorkie (Yki) to the nucleus, which transcriptionally activates the expression of *vg* through its quadrant enhancer (*vg*QE). Finally, Vg expression in the newly-recruited cell closes the loop by transcriptionally inhibiting *ds* expression and thereby fueling the propagation of Ds-Ft polarity. Thus, *ds* expression within the wing pouch defines the population of potentially recruitable cells. The recruitment of cells into the Vg pattern continues until completion of the whole wing disc pouch [31]. In 2020, Muñoz-Nava *et al.,* studied the contribution of cell recruitment to normal wing size and showed that adult wings are on average 21% smaller

when cell recruitment is impaired [32]. However, no studies have been conducted to investigate the effects of cell recruitment on wing shape.

Here, we hypothesize that cell recruitment contributes to AP asymmetries in the *Drosophila* wing disc. Using morphometrics methods [33, 34], we first show that lack of recruitment does result in a more symmetric wing by exhibiting a reduction of the area comprised between vein L5 and the wing margin M (L5-M). Contrary to expectations, the reduction of the L5-M domain cannot be explained by a similar reduction in the corresponding pro-vein region during larval stages. However, impairment of cell recruitment does generates a change in the circularity of the *vg* pattern. Thus, our work suggests that cell recruitment contributes to the establishment of AP asymmetries in this system by changing the geometry of the Vg pattern.

## Materials and methods

### Fly stocks and crosses

The following stocks were used: (1) *y,w* provided by Fanis Missirlis (Cinvestav, Mexico), (2) UAS-*vg*RNAi (Vienna Drosophila Resource Center # 16896), (3) *ds*-Gal4,UAS-GFP/CyO; vgQELacZ/[TM6B,Tb] provided by Gary Struhl, (Columbia University, New York, USA), (4) *vg*QE-Gal4 (Bloomington Drosophila Stock Center [BDSC] # 8229), (5) *y,w*, hs-FLP; Act>CD2>Gal4, UAS-RFP (BDSC # 51308). All fly crosses were conducted at 25˚C, unless otherwise specified. We conducted crosses to obtain the following balanced stocks: (i) (*ds*-Gal4, UAS-GFP;MKRS)/(SM5;TM6B, Tb), (ii) *y,w*; UAS-*vg*RNAi; *vg*QELacZ/TM6B, and (iii) *y,w*, hs-FLP; Act>CD2>Gal4, UAS-RFP/ UAS-VgRNAi. To genetically block cell recruitment, we crossed (*ds*-Gal4, UAS-GFP;MKRS)/(SM5; TM6B, Tb) males with *y,w*; UAS-*vg*RNAi; *vg*QELacZ/TM6B females and selected animals of the phenotype *ds*-Gal4, UAS-GFP/ UAS-*vg*RNAi; MKRS/*vg*QELacZ. These animals express a *vg*RNAi in the pattern of *ds*, which is located at the periphery of the wing pouch, including cells that have not yet expressed Vg through recruitment ([32]; S1 Fig). As a control experiment, we crossed *y,w* males to *y,w*; UAS *vg*RNAi; *vg*QELacZ/[TM6B, Tb] females to obtain UAS-*vg*RNAi/+; *vg*QELacZ/+. Since RNAi lines often have variable effects, we tested that this line exhibits a very strong reduction of Vg levels and function in mosaics of *y,w*, hs-FLP; Act>CD2>Gal4, UAS-RFP/ UAS-*vg*RNAi animals (S2 Fig). *vg*RNAi expression was induced in mosaics by a 5 min heat shock at 37˚C followed by a 48 h growth time before dissection (S2 Fig).

### Wing imaginal disc dissection and immunostaining

Wing imaginal discs were dissected from third-instar larvae of either sex under a stereoscopic microscope using PEM buffer: 80 mM Na-Pipes (Sigma-Aldrich, cat. # P2949), 5 mM EGTA (Sigma-Aldrich, cat. # E-3889) and 1 mM $MgCl_2$ x 6 $H_2O$ (Sigma-Aldrich, cat. # 2555777), pH 7.4. We use a standard immunostaining protocol originally obtained from E. Bier (UCSD).

Discs were fixed in PEM-T (PEM with 0.1% of Triton X-100, Sigma-Aldrich, cat. # T9284) with 4% paraformaldehyde for 35 minutes, washed 3 times and blocked in PEM-T with 0.5% of Bovine Serum Albumin for 2 h at room temperature. Then, samples were stained with primary antibodies at 4˚C overnight at the following dilutions: monoclonal mouse anti-DSRF (*Drosophila* Serum Response Factor) a gift from S. Blair (1:250) and rabbit *β*-Galactosidase (MP Biomedicals, cat. # 55976, 1:250). Primary antibodies were detected with Alexa Fluor 488 anti-mouse and anti-rabbit Alexa Fluor 647 secundary antibodies at 1:1000. Imaging was conducted in a Leica TC5 SP8 confocal microscope using a 40X oil-immersion objective.

## Analysis of fluorescence patterns in wing discs

Z-stacks of confocal images were loaded in Python using the readlif library [35]. For analysis of *vg*QELacZ patterns, we took the Z-projection and deleted noise using the morphology function of Scikit-image. We binarized the images using the threshold function of OpenCV at 20% of the maximum pixel value in the pattern. To obtain the pro-veins patterns in the wing disc, we took the Z-projection of the DSRF staining, a typical intervein marker [36], and obtained an inverse binarization of this image using the threshold function of OpenCV. Then, we manually clicked points between DSRF intervein pattern, in order to determine pro-vein patterns, and saved the xy coordinates in a numpy array. Then, we binarized the *vg*LacZ pattern using the threshold function from OpenCV and calculated the total area of the binarized pattern. We used the vein coordinates to segment the *vg*LacZ pattern into polygons corresponding to the intervein regions M-L2 (from the anterior edge of the *vg*LacZ pattern to the coordinates of vein 2), L2-L3, L3-L4, L4-L5 (the regions of the *vg*LacZ pattern between the coordinates of the corresponding veins), and the L5-M region (from the coordinates of vein L5 to the posterior edge of the *vg*LacZ pattern). The area of each section was computed with the contourArea function from openCV. The relative area was calculated as the ratio between the area of the section and the total area of the *vg*LacZ pattern. Alternatively, we also measured the area of intervein sections in a Vg-independent manner by considering the wing pouch as the ellipse passing through the hinge-pouch (HP) folds (S3 Fig). To obtain the points on the HP folds, we manually clicked on the points in the disc immunostained with DAPI. We then used the EllipseModel function from skimage.measure to fit, by least squares, the ellipse passing through the clicked points and used the coordinates of the polygons between the veins and the fitted ellipse. The area of each section was computed as the area of the corresponding polygon, using the contourArea function from OpenCV. The relative area was calculated by dividing the area of the section by the total area of the ellipse. We analyzed 36 control wing discs and 16 recruitment-impaired wing discs.

To measure the eccentricity of the Vg pattern, we binarized the *vg*QELacZ pattern, smoothed the edges using OpenCV's blur function, and computed the convex envelope of the pattern using the ConvexHull function of Python's scipy.spatial package. The convex hull of the *vg*QELacZ pattern is the smallest set of points that form a convex polygon (*i.e.*, no diagonal between two internal points lies outside the polygon) that can enclose the entire *vg*QELacZ pattern. We then used the points of the convex hull to fit an ellipse using the EllipseModel function from skimage.measure. The eccentricity was calculated using the formula

$$e = \sqrt{1 - \frac{b^2}{a^2}}$$

where $e$ is the eccentricity, $a$ is the semi-major axis length, and $b$ is the semi-minor axis length. $a$ and $b$ were computed by the EllipseModel function. We analyzed the same discs used to measure the intervein areas, *i.e.*, 36 control wing discs and 16 experimental wing discs.

To analyze the pixels where *ds* overlaps with *vg* (S1 Fig), we examined control discs that activate GFP driven by *ds*, using the Gal4-UAS system, and immunostained for *vg*QELacZ. We analyzed discs during the early third instar (ETI) and the late third instar (LTI) larval stages. We averaged the pixel values in a 30-pixel square region outside the pattern of interest (e.g., in the central region of the pouch for *ds*, or in the notum region of the wing disc for *vg*). We then subtracted the average from the image of interest to remove background noise. We then obtained the maximum projection every 5 stacks in the GFP and *vg*LacZ channels and quantified the number of positive pixels for GFP and *vg*LacZ. Additionally, we estimated the area of a nucleus in our images, which we determined to be 200 pixels, and then divided the

number of overlapping pixels by 200 to estimate the number of nuclei overlapping in both patterns. We analyzed 4 ETI and 6 LTI wing discs.

## Wing mounting and wing area quantification

Adults wings were mounted as previously reported in [32]. Briefly, we sorted adult flies by sex and then dehydrated them overnight in 70% ethanol. Wings were then dissected in 50% ethanol and mounted in a glass coverslip. We imaged adult wings in a Nikkon bright-field microscope using a 4X objective. We determined the border between wing blade and hinge using a straight line from the proximal border of the costa to the proximal border of the alula. Then, we measured the total wing area using the polygon selection tool in ImageJ. Subsequently, we quantified the areas between veins L1-L2, L2-L3, L3-L4 (ignoring the anterior crossvein), L4-L5 (ignoring the posterior crossvein), and the area between vein L5 and the lower wing margin (L5-M) using the polygon selection tool in ImageJ. The normalized area of each region was calculated by dividing the area of the region by the total wing area. We analyzed 52 control and 20 recruitment-impaired wings from male flies, and 58 control and 21 recruitment-impaired wings from female flies. We did not discriminate between right and left wings.

## Procrustes analysis

Images of adult wings were loaded in Python using the OpenCV library. We used the scipy. procrustes package for Procrustes analysis. The procrustes function receives two matrices that contain the reference points (landmarks) of the shapes to be compared. In this case, the control landmarks and the experimental landmarks. It returns the matrices mtx1, mtx2, and the disparity value. mtx1 contains the coordinates of the control landmarks, adjusted in scale, rotation, and translation to optimize their correspondence with the experimental points. mtx2 contains the coordinates of the experimental landmarks, similarly transformed and aligned with the control reference points. The disparity value is a measure of the difference between the two configurations of points after the transformations, calculated as the sum of the squared distances between corresponding points.

The margin landmarks (ML) consists of 10 points, 6 of which are common in prior *Drosophila* wing Procrustes analysis [12, 37]. The other 4 points are obtained by extending the lines defined by the anterior and posterior cross veins and finding the intersection of these lines with the wing margin. The vein landmarks (VL) consists of 10 points located over the venation pattern, as in prior studies [12, 37]. To compare the ML and VL, we analyzed the same control (n = 52) and recruitment-impaired (n = 20) wings that were used to quantify the areas of the intervein regions. We performed a comprehensive pairwise comparison, where each of the 52 male control wings was compared against each of the male 20 recruitment-impaired wings. This resulted in a total of $52 \times 20 = 1040$ Procrustes distance measurements. To calculate the average landmarks, we averaged the matrices mtx1 and mtx2 for each Procrustes comparison. An equivalent analysis was conducted with the 58 female control and 21 female recruitment impaired wings.

For the symmetrization analysis of Fig 2, we obtained the mid-point between points 2A and 2P (Fig 2A) and the mid-point between points 5A and 5P. The AP axis is hereby defined as the line through these two mid-points. The landmarks to quantify symmetrization consists of 10 points common in the prior *Drosophila* wing Procrustes analyses [12, 37], such that for each point there is a corresponding counterpart with respect to the AP axis defined above. Relative to this axis, vein L2 was considered the counterpart of vein L5, and vein L3 the counterpart of vein L4. To obtain the anteriorized wing, points 1A-5A were reflected with respect to the AP axis. To carry out the reflection, the straight line orthogonal to the AP axis that passed through

each point of interest (1A-5A) was obtained. The distance from the line to the point was measured, and on the orthogonal line, in the posterior region, at the same distance as the point of interest, the symmetrical point was marked. For the posteriorized wing, an equivalent process was carried out, reflecting points 1P-5P with respect to the AP axis. These analysis were performed in a homemade Python code (S5 File). Similar to the ML and VL comparison, we analyzed the same control (n = 52) and recruitment-impaired wings (n = 20) that were previously used to quantify intervein areas. For the comparison between control and anteriorized control wings, we calculated the Procrustes distance of each wing with its symmetrized counterpart, resulting in 52 Procrustes distance measurements. We conducted similar analyses for the comparisons between control wings and posteriorized control wings, recruitment-impaired wings and anteriorized recruitment-impaired wings, and recruitment-impaired wings and posteriorized recruitment-impaired wings. For the recruitment-impaired wings, this resulted in 20 Procrustes distance measurements for each comparison. An equivalent analysis was conducted with 58 female control wings and 21 female non-recruitment wings.

## Statistical analysis

In all comparisons of two data sets (control vs. experimental) were compared, the following procedure was carried out: A normality test was conducted for each data set using the Shapiro-Wilk test using the normality function from the pingouin package in Python. The null hypothesis is that the data follow a normal distribution. A significance level of $\alpha = 0.05$ was considered. In the cases where the normal distribution hypothesis was not rejected, an equality of variances test was conducted using the levene function from the scipy.stats library in Python. Subsequently, a Student's t-test was performed using the ttest_ind function from scipy.stats, specifying equal_var = True in the case of equal variances ($p - value > 0.05$ in the Levene test) or equal_var = False in the case of unequal variances ($p - value < 0.05$ in the Levene test). Conversely, if at least one of the groups did not follow a normal distribution, a non-parametric Mann-Whitney U test was conducted using the mwu function from the pingouin package in Python, considering a significance level of $\alpha = 0.05$. We then conducted statistical power tests using the TTestIndPower class from the statsmodels package in Python for the t-tests, or using a homemade Python function (S5 File) specifically for calculating the power of the Mann-Whitney U test, employing a Monte Carlo simulation. The results of statistical computations are provided in S1–S4 Files.

## Results

### The inhibition of cell recruitment mainly affects the L5-M region and wing shape

In order to investigate how cell recruitment affects wing shape, we employed flies in which Vg expression was impaired in the pattern of *ds*, which is complementary to the *vg* pattern and thereby blocks that Vg expands further to the rest of the wing pouch (*ds*-Gal4, UAS-*vg*RNAi; [32]; S1 Fig). In particular, we compared the area of the intervein sections of the wing (L1-L2, L2-L3, L3-L4, and L5-M; Fig 1A) in control and *ds*-Galg4,UAS-*vg*RNAi flies. We found that most of these regions (L1-L2, L2-L3, L3-L4) marginally increased their relative size in recruitment-impaired wings with respect to controls (Fig 1B). In contrast, the L5-M region decreased its relative size in 15% (Fig 1B). This effect does not depend of sex, since a similar result is obtained for male and female individuals (S4A Fig). These results suggest that cell recruitment contributes to growth of the L5-M region.

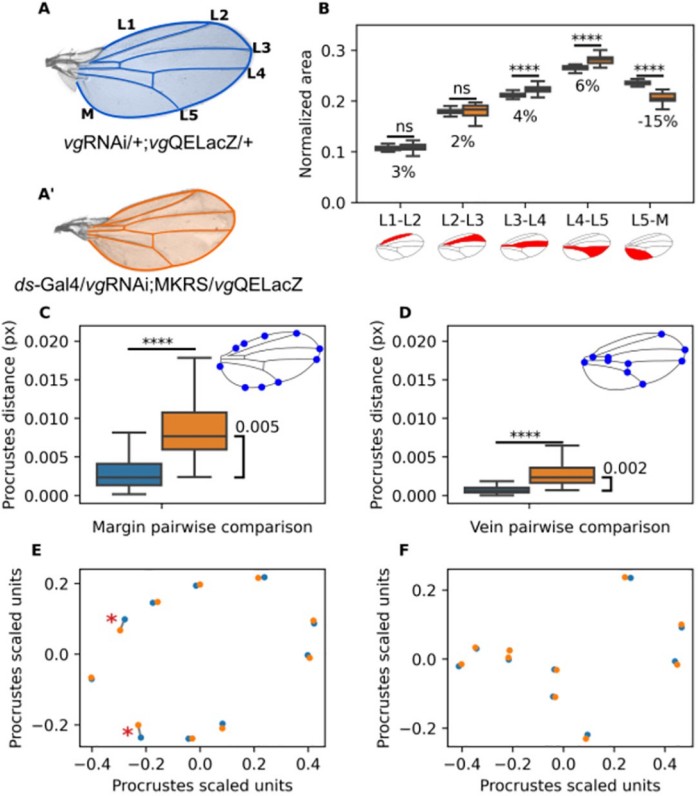

**Fig 1. Genetic impairment of cell recruitment affects the area of the region L5-M and wing shape. A,A'.**
Representative control (A) and recruitment-impaired (A') wing showing the veins (L1-L5) and the Margin (M). In blue shading we show the wing blade, while blue lines highlight the vein pattern. **B**. Normalized area of each intervein region with respect to the total area of the wing. Color-coding is as in A. Numbers under the boxplot represent the percentage of increase in the medians between the groups. **C,D.** Procrustes comparison of the Margin landmarks (C) and Vein landmarks (D). Blue: Control vs. Control. Orange: Control vs. Recruitment-impaired. In the upper left corner of both plots, representative images of ML (C) and VL (D) are shown. **E,F**. Average displacement of control landmarks (blue points) and recruitment-impaired landmarks (orange points) after Procrustes analysis for the ML (E) and for the VL (F). The asterisks indicate the landmarks that show greater displacement between control and recruitment-impaired conditions. Sample sizes: Control wings (n = 52); Recruitment-impaired wings (n = 20). All wings in this figure were dissected from male flies. For the analysis using female wings see S4A–S4C Fig. A Shapiro test shows that distributions are non-parametric. Thus, A Mann-Whitney U test was conducted. * indicates $p < 0.05$ and **** indicates $p < 5 \times 10^{-5}$ (see S1 File for statistical computations).

In order to quantitatively measure wing shape, we used Procrustes analysis (a metric of shape changes which is scale, rotation, and translation invariant [33, 34]) using two different sets of landmarks defined on the wing. The first, hereby referred as the Margin Landmarks (ML, Fig 1C), is comprised of points on the wing margin and reflects overall wing shape. The second, hereby referred as the Vein Landmarks (VL, Fig 1D), is comprised of points on the wing veins and reflects the venation pattern. We compared recruitment-impaired wings and control wings both using ML and VL using the Procrustes distance. As an basal comparison, we computed Procrustes distance between each pair of control wings (blue bars in Fig 1C and 1D) and between each pair of control *vs.* recruitment-impaired wings (orange bars in Fig 1C and 1D). We found significant differences in Procrustes distance between control *vs.* recruitment-impaired comparisons and control *vs.* control comparisons, both using ML (Fig 1C) and VL (Fig 1D), but the difference in Procrustes distance is 2.5 times larger in the ML than in the VL comparisons (Fig 1C and 1D). In order to determine which landmarks are the most

affected, we compared the coordinates of each landmark in recruitment-impaired wings for both the margin and the venation sets of landmarks (Fig 1E and 1F). We found that the most of the landmarks are displaced similarly along both the margin and the venation pattern. However, two landmarks located in the proximal part of the margin displaced twice more than the average (asterisks in Fig 1E). Particularly, the one located in the anterior margin appears to be displaced along the margin, and has little effect in the shape of the wing; however, the one located in the posterior margin is displaced in a direction orthogonal to the margin and it likely explains the reduction of the L5-M area in recruitment-impaired wings. We conclude that the lack of recruitment remarkably change wing shape but has little impact on the venation pattern.

## Recruitment-impaired wings are more symmetric along the AP axis compared to controls

Our previous result shows that cell recruitment appears to contribute more to growth of the posterior region of the wing (particularly the L5-M area, Fig 1B). We then asked if cell recruitment is indeed a developmental mechanism that contributes to AP symmetry-breaking in the *Drosophila* wing. Namely, we hypothesize that recruitment-impaired wings are more symmetric than control wings. To test this, we compared control and recruitment-impaired wings to 'symmetrized' versions of themselves. In particular, we define the Anteriorized (Posteriorized) image of a particular wing (Fig 2A) as the mirror-image reflection of the anterior (posterior) part of the wing with respect to the AP axis (see Materials and methods; Fig 2A). Then, for every wing we can obtain a measure of AP symmetry by computing the Procrustes distance to the Anteriorized and Posteriorized images of itself (Fig 2A). We found that recruitment-impaired wings are on average closer (in Procrustes distance) to both, their Anteriorized and Posteriorized images, compared to control wings (Fig 2B). This result is even more evident when a more extensive set of landmarks is used (see S5 Fig). We conclude that recruitment-impaired wings exhibit more AP symmetry than controls.

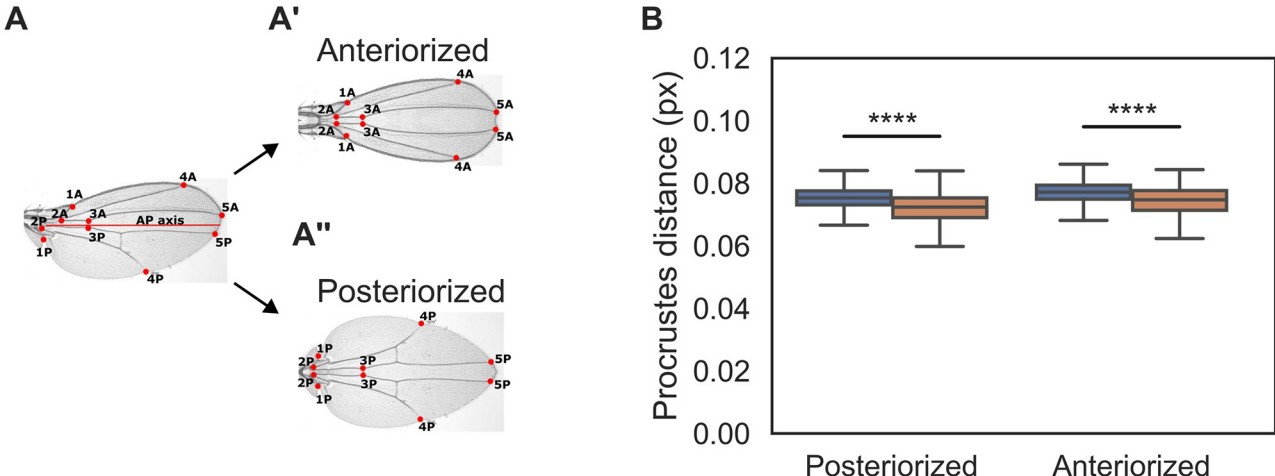

**Fig 2. Recruitment-impaired wings are more symmetric compared to controls. A.** Representative control wing along with its Anteriorized and Posteriorized images. We identified a set of 10 vein landmarks and labeled them such that for each landmark in the Anterior (A) side, there is another landmark in the posterior (P) side (see Materials and methods). This defines 5 pairs of landmarks in each wing (left). Hypothetical wing images constructed in this way are mirror-image duplications of their anterior (A') and posterior (A") parts with respect to AP axis. **B.** Comparison of the control wings (blue bars) vs. recruitment-impaired (orange bars) with their respective posteriorized or anteriorized wings. Sample size for each set is as in Fig 1. An equivalent analysis of wings from female flies is shown in S4D Fig. A Shapiro test shows that distributions are non-parametric. Thus, a Mann-Whitney U test was conducted. **** indicates $p < 5 \times 10^{-5}$ (see S2 File for statistical computations).

### Impairment of cell recruitment affect the circularity of the wing pouch but does not affect the L5-M intervein area of the *Drosophila* wing disc

In order to test if cell recruitment contributes to breaking the AP asymmetry of the *Drosophila* wing by recruiting more cells in the pro-vein domain that will give rise to the L5-M region of the adult wing, we examined the consequences of impairing recruitment in larval development. We analyzed a reporter of *vg*QE (*vg*QELacZ) in the third instar of the *Drosophila* wing disc (Fig 3A' and 3B'). We stained these discs with antibody of DSRF (Fig 3A and 3B), a specific marker for the intervein regions (Blair, 2007), in order to measure each intervein area (Fig 3A" and 3B"). Contrary to our expectations, we did not find significant differences between the relative areas of intervein regions in controls and recruitment-impaired discs (Fig 3C). Due to the difficulty in determining the lateral margins of the wing pouch, we also analyzed the ellipse passing through the Hing-Pouch (HP) folds, another typical marker of the wing pouch, to quantify the intervein sections. However, again, we did not find significant differences in the relative size of the L5-M region when comparing between control and recruitment-impaired wing discs (S3 Fig).

We also investigated if blocking cell recruitment has an effect on the overall shape of the wing pouch. To check this, we fitted an ellipse which passes through the contour of the *vg*QE-LacZ pattern (see Materials and methods, Fig 4). We measured the eccentricity of the fitted ellipses, and found that in recruitment-impaired discs, the wing pouch is more circular compared to control (*i.e.,* the eccentricity of the fitted ellipse is smaller, Fig 4B). Taken together,

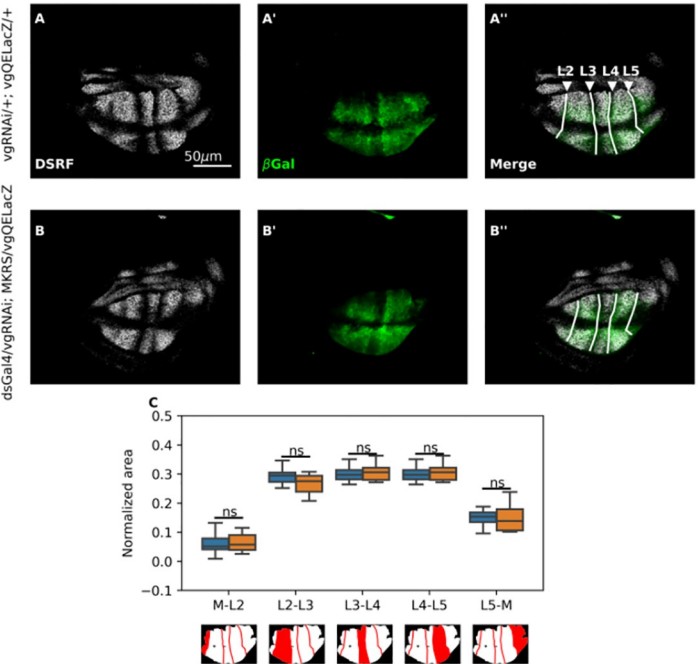

**Fig 3. The reduction of the L5-M area in recruitment-impaired adult wings can not observed during larval development. A-B'.** Representative control (A, A') and recruitment-impaired third-instar wing discs (B, B') immunostained with DSRF (A, B, white) and *β*-galactosidase (A', B', green) antibodies. Arrowheads in A" indicate the position of the veins. A" and B" show the merge of these patterns. All discs carry the *vg*QELacZ reporter. **C.** Normalized area with respect to the total area of the *vg*QELacZ pattern of each intervein region (see Materials and methods). Diagrams below this graph illustrate the intervein regions in the wing disc. Sample sizes: Control wings (n = 36); Recruitment-impaired wings (n = 16). A Shapiro test shows that distributions are non-parametric. Thus, a Mann-Whitney U test was conducted. ns indicates that $p > 0.05$ (see S3 File for statistical computations).

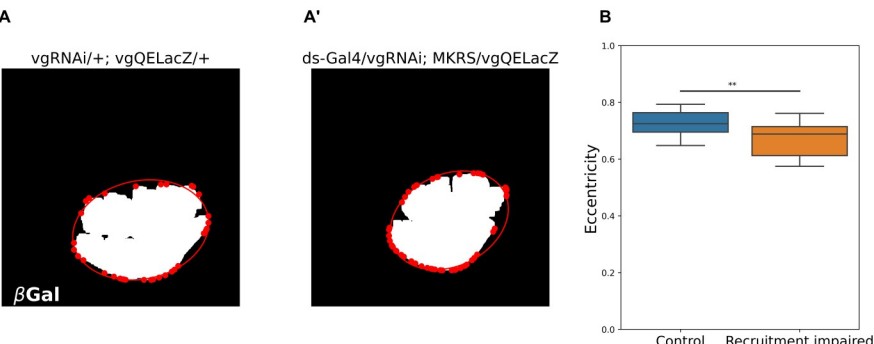

**Fig 4. The pattern of *vg* QELacZ in recruitment-impaired discs is more circular than in controls. A-A'.**
Representative control (A) and recruitment-impaired (A') wing discs showing their binarized *vg*QELacZ patterns. The
representative discs used here are the same as in Fig 3A' and 3B'. **B**. Eccentricity of the ellipse fitted to the *vg*QELacZ
pattern using a convex envelope analysis of the pattern (see Materials and methods). Sample size for each set is as in
Fig 3. A Shapiro test shows that distributions are non-parametric. Thus, a Mann-Whitney U test was conducted. **
indicates $p < 0.01$ (see S4 File for statistical computations).

our data and analysis suggest that cell recruitment does contribute to the overall shape of the
the wing pouch but cannot explain why the L5-M region is smaller in adult wings.

## Discussion

The developmental mechanisms underlying organ shape have been of fundamental interest in
biology since the publication of D'Arcy Thompson's seminal work 'On Growth and Form'
more than 100 years ago [38]. This gave rise to the field of morphometrics, the quantitative
analysis of shapes, which has advanced several disciplines, from the integration of ecology,
evolution, and development (eco-evo-devo) to human evolution [39–44]. However, the pro-
cesses orchestrating the establishment of an organ's final form remain elusive. In the search for
developmental effectors underlying tissue shape, previous research has predominantly focused
on examining both local and global forces acting on tissues [9, 10]. On the other hand, signal-
ing pathways influencing early development have been linked to positional information that
confers identity to organs or segments [45, 46]. In addition, the attainment of the final tissue
shapes has been associated with cell movement and adherence to the extracellular matrix [9].
Despite these efforts, significant gaps persist in integrating early signaling, growth, cell adhe-
sion, and mechanical processes underlying the developmental basis of organ shape. One par-
ticular aspect that remains poorly understood in this broad problem is uncovering the origins
of organ asymmetries.

The *Drosophila* wing serves as an excellent model for studying the genetics basis of organ
growth and shape [47]. Early studies in the *Drosophila* wing disc have revealed the role of
*engrailed/invected* and *apterous* in establishing the distinction of AP and DV compartments,
respectively [36, 48, 49]. Strikingly, while DV patterning in the developing wing requires to be
symmetric in order to ensure that dorsal and ventral sides of the adult wings match, AP pat-
terning is inherently asymmetric, as revealed by clear differences in the positions of veins and
intervein areas (Fig 1). Morever, Muñoz-Muñoz et al. (2016) previously showed that in
response to environmental perturbations, genetic variations (between isogenetic lines), and
right-left fluctuations, the *Drosophila* wing exhibits a Proximal-Distal (PD) compartmentaliza-
tion but does not exhibit an AP compartmentalization [50]. The obvious suspect of the AP
asymmetries in the *Drosophila* wing is Dpp signaling, which is indeed asymmetric due to

receptor activity, leading to asymmetric pMad expression [51]. Yet, this asymmetry does not explain why the L1-L2 and L5-M intervein areas are remarkably different. Moreover Matsuda *et al.*, have revealed that the range of the AP signal is confined to the territory comprised between L2 and L5 [20]. Therefore, it is unlikely that Dpp signaling could explain the asymmetry of these intervein areas [20]. Here, we inhibited cell recruitment, a mechanism that drive the expansion of *vg* through the activation of the QE. Our findings indicate that in recruitment-impaired wings, L5-M intervein area is significantly reduced (Fig 1B). This reduction is not comparable to the relative size of the L1-L2 intervein area (Fig 1B), suggesting that while cell recruitment could contribute to this asymmetry, it cannot fully explain it and other mechanisms must act to redundantly establish AP asymmetries in the wing.

In light of the work of Matsuda *et al.,* (2020), where it has been shown that the range of the Dpp signal is not required for the growth of the compartment beyond L5, we were surprised to find that inhibition of recruitment did not result in a significant reduction of the L5-M domain in the wing disc (Fig 3C). However, we noted that while the adult wing margin has a clear border, there is not such a clear-cut end of the *vg*QE pattern in the lateral regions of the disc. Furthermore, the pattern that determines the intervein domains, marked by DSRF, overlaps exactly in the territories within veins L2-L5, but differs in lateral regions, suggesting that *vg*QE and DSRF expression could establish different territories at the edges of the pouch (Fig 3). Indeed, *vg* expression possibly contributes to the establishment of the wing appendages, called costa and alula [52], so to draw a clear boundary between the blade and these structures is not obvious.

Another aspect that deserves some discussion is the potential role of cell recruitment in tissue mechanics. Since the wing disc is a pseudo-stratified tissue [22], the tensions generated by cell-cell anchoring depend largely on the number of cells that make up the tissue [25]. Therefore, reducing the number of cells in a specific domain of the wing disc can generate a regression of the entire tissue during pupal stages. Examining changes in these regions during pupal stages is challenging and is beyond the scope of this paper. However, the change in the circularity of the *vg*QE pattern, upon impairment of cell recruitment, could explain geometrically the reduction of the L5-M area and a slight increase of more central areas (Fig 1B), simply because making and elliptic pattern more rounded reduces the difference between its major and minor axes, which results in making areas at more curved edges smaller at the expense of making central regions larger.

Our work reveals for the first time that cell recruitment in the *Drosophila* wing could have an impact on the overall asymmetric shape of adult wings. Therefore, in addition to its previously characterized role in patterning and growth control [32, 53, 54], cell recruitment may also contribute to establishing asymmetric shape changes during development.

## Supporting information

**S1 Fig. The Ds pattern does not overlap with the Vg pattern during the third larval instar.**
**A,B.** Control wing disc expressing GFP driven by *ds* using the Gal4-UAS system during early third instar (ETI) (A) and late third instar (LTI) (B), immunostained with *β*-galactosidase (A', B'). A'' and B'' show the merge of these patterns. All discs carry the *vg*QE-LacZ reporter.
**C.** Approximate number of nuclei that simultaneously express *vg* and *ds* at each of the stages analyzed (see Materials and methods). Sample size: ETI wing discs (n = 4); LTI wing discs (n = 6).
(TIF)

**S2 Fig. Verification of RNAi efficiency through the expression of a *vg* RNAi in mosaics.**
**A-B.** Generation of genetic mosaics expressing the red fluorescent protein (RFP) through the

Gal4-UAS system activated by the FLP-FRT system after heat shock (A) and coexpression of RFP along with *vg*RNAi (B). **A'-B'** Immunostaining with a Vg antibody in the wing disc pouch (A'). Arrowheads indicate a cell nuclei expressing RFP, but lacking Vg expression due to RNA interference action. **A"-B"** Merge of these patterns. Sample size, n = 9.
(TIF)

**S3 Fig. The ellipse that passes through the HP folds and delimitates the wing pouch can not recapitulate the reduction of the L5-M region in recruitment-impaired animals. A-C.** Representative control wing disc immunostained with DAPI (A,B) and DSRF (C). In A, we mark the position of the HP folds with arrowheads. In B, we mark some points, that pass through the HP fold and were used to fit, by least squares, the ellipse shown in red. In C, the veins are marked with red dashed lines and arrowheads indicate the position of the L2-L5 veins. **D**. Overlay of the fitted ellipse with the veins to quantify the areas of the intervein sections. The diagrams below this graph illustrate the intervein region analyzed in the wing disc. Sample sizes: control wing disc (n = 30); recruitment-impaired wing disc (n = 16). ns indicates $p > 0.05$.
(TIF)

**S4 Fig. Impairment of cell recruitment affects mainly the L5-M area and wing shape in females. A.** Normalized area of each inter-vein region with respect to the total area of the wing. Color coding and analysis is as in Fig 1. **B-C.** Procrustes comparison of the ML (B) and VL (C). **D**. Procrustes comparison for Anteriorized and Posteriorized control and recruitment-impaired wings. Color coding and additional information is as in Fig 2. Sample sizes: control wings (n = 58), recruitment-impaired wings (n = 21). A Shapiro test shows that distributions are non-parametric. Thus, a Mann-Whitney U test was conducted. * indicates $p < 0.05$, *** indicates $p < 0.001$ and **** indicates $p < 0.00005$.
(TIF)

**S5 Fig. Recruitment-impaired wings exhibit even more AP symmetry when the set of landmarks is more extensive.** Left. A more comprehensive set of landmarks comprised of 26 points. Right. Procrustes comparison each control and recruitment-impaired wings with respect to their Anteriorized and Posteriorized images using the landmarks shown on the left. Color code is as in Fig 2B. Sample size for each set is as in Fig 2. A Shapiro test shows that distributions are non-parametric. Thus, a Mann-Whitney test was conducted. **** indicates $p < 5 \times 10^{-5}$.
(TIF)

**S1 File. Statistical analysis of the data in Fig 1.**
(CSV)

**S2 File. Statistical analysis of the data in Fig 2.**
(CSV)

**S3 File. Statistical analysis of the data in Fig 3.**
(CSV)

**S4 File. Statistical analysis of the data in Fig 4.**
(CSV)

**S5 File. Python code to compute the statistical power of the Mann-Whitney U test.**
(PY)

## Acknowledgments

We thank members of the Nahmad laboratory for discussions, Jose Luis Fernandez for technical support, and Seth Blair for sharing an aliquot of the DSRF antibody.

## Author Contributions

**Conceptualization:** Rosalío Reyes, Marcos Nahmad.

**Data curation:** Rosalío Reyes, Rafael Rodriguez-Muñoz.

**Formal analysis:** Rosalío Reyes.

**Investigation:** Rosalío Reyes, Marcos Nahmad.

**Methodology:** Rosalío Reyes, Rafael Rodriguez-Muñoz.

**Project administration:** Marcos Nahmad.

**Software:** Rosalío Reyes.

**Supervision:** Marcos Nahmad.

**Validation:** Rosalío Reyes, Marcos Nahmad.

**Visualization:** Rosalío Reyes.

**Writing – original draft:** Rosalío Reyes, Marcos Nahmad.

**Writing – review & editing:** Rafael Rodriguez-Muñoz.

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
