## [Decision Letter · Decision Letter 0]

3 Jul 2024

PONE-D-24-23042Cell recruitment and the origin of Anterior-Posterior asymmetry in the Drosophila wingPLOS ONE

Dear Dr. Nahmad,

Thank you for submitting your manuscript to PLOS ONE. After careful consideration, we feel that it has merit but does not fully meet PLOS ONE’s publication criteria as it currently stands. Therefore, we invite you to submit a revised version of the manuscript that addresses the points raised during the review process.

The reviews for your manuscript have been received. The reviewers' comments are included along with this letter. Independent reviewers have assessed the manuscript's suitability for publication and solicit additional clarification on certain outstanding issues. Based on the reports of the reviewers and my assessment, I recommend that you revise the manuscript to address the concerns raised.

If you wish to revise the manuscript to address the issues raised by reviewers, I would be happy to consider a revised version of the manuscript for publication. When revising the manuscript, please consider all the points raised by the reviewers and outline every change made. If you disagree with the reviewers' comments, you should provide a suitable rebuttal to their concerns. 

We look forward to receiving your revised manuscript.

Kind regards,

Abhinava Kumar Mishra, PhD

Academic Editor

PLOS ONE

Journal Requirements:

4. Please remove your figures from within your manuscript file, leaving only the individual TIFF/EPS image files, uploaded separately. These will be automatically included in the reviewers’ PDF.

Reviewers' comments:

Reviewer's Responses to Questions

**Comments to the Author**

1. Is the manuscript technically sound, and do the data support the conclusions?

Reviewer #1: Partly

Reviewer #2: Yes

Reviewer #3: Partly

Reviewer #4: Partly

Reviewer #5: Yes

Reviewer #6: Yes

Reviewer #7: Yes

2. Has the statistical analysis been performed appropriately and rigorously? 

Reviewer #1: Yes

Reviewer #2: Yes

Reviewer #3: Yes

Reviewer #4: Yes

Reviewer #5: No

Reviewer #6: Yes

Reviewer #7: Yes

3. Have the authors made all data underlying the findings in their manuscript fully available?

Reviewer #1: Yes

Reviewer #2: Yes

Reviewer #3: Yes

Reviewer #4: Yes

Reviewer #5: Yes

Reviewer #6: Yes

Reviewer #7: Yes

4. Is the manuscript presented in an intelligible fashion and written in standard English?

Reviewer #1: Yes

Reviewer #2: Yes

Reviewer #3: Yes

Reviewer #4: Yes

Reviewer #5: No

Reviewer #6: Yes

Reviewer #7: Yes

5. Review Comments to the Author

Reviewer #1: Reyes et al. describe an interesting phenomenon that knockdown of vg results in asymmetric growth of the adult wing in Drosophila. It shows a great decrease of the L5-M region, whereas other areas of the wing are increased. However, the authors did not recapitulate this phenotype in the larval stage. The authors presume that the overall asymmetry of the adult wing may be attributed to, at least partially, cell recruitment.

However, essential data, such as cell proliferation, is missing to make the conclusion reliable. Below are several comments and criticisms that need to be addressed.

1. In Materials and Methods, the authors would clarify:

(1) To show what PEM stands for when it first appears in the text.

(2) How to quantify the relative area?

(3) To show the number of wings for statistical analysis.

2. What's the expression pattern of dsGal4 in the wing disc? I hope the authors use only dsGal4 or ds-Gal4 in the whole manuscript.

3. Label in Fig. 3A or B which line represents L1-L5 and M, respectively, in the larval wing disc.

4. The authors found no obvious defects in the L5-M region in the larval stages. What about the area in the pupal stage, such as 32 h after pupation when vein fate induction is conspicuous in the pupal wing (Johannes and Preiss, 2002). What happens to the cell proliferation rate during late third to early pupal stages.

5. In Matsuda et al. (2021) work, “it has been shown that the range of the Dpp signal is not required for the growth of the compartment beyond L5”, but there are other works to show that the Dpp signaling promotes cell proliferation in the lateral region of the wing disc (Zhang et al. 2013, Development 140, 2917-2922). The authors would reorganize this conclusion.

6. To discuss the relevance of more circular phenomenon as shown in Fig. 4 with the reduction of L5-M area, and the increase of other areas?

Reviewer #2: In this manuscript, Rosal´ıo Reyes et. al. attempted to figure out the roles of “Vg-cell recruitment”, a non-cell-autonomous autoregulation of Vg in the global shaping of Drosophila wing. The authors have previously established the ds-Gal4 > UAS-vg RNAi system which impaired the cell recruitment and found cell-recruitment is required for the wing to be a complete size. Here, through this “recruitment-impairment fly” the authors showed that cell recruitment is required for the A-P asymmetric shaping of wing. The concept that the cell recruitment did not only regulate tissue growth but tissue shaping is very interesting. However, several interpretations in the experiments do not seem to be logical. I consider that several additional experiments are needed to confirm the authors’ argument.

Major comments

1. Flies with vg RNAi always have MKRS chromosome in their background while control flies do not have it. I concern that Minute mutation (M(3)76A) included MKRS may affect the wing development. The authors should remove MKRS.

2. The photos showing complementary expression of the ds-Gal4 and vg expression (vg-lacZ) in the developing wings are needed (e.g. L2, L3 and wandering L3 stage). I concern that geometrical changing of wing pouch may not be due to the cell-recruitment impairment but the local suppression of vg shaped by the unexpectedly overlapped-ds-Gal4 pattern in the distal region of the wing pouch.

3. (Fig. 1D) The authors should specify the landmarks that exhibit the significant geometrical changing by the cell-recruitment impairment.

4. (line 183, Fig. 2B) The authors should explain or discuss what causes the wing shape symmetrically in the recruitment impairment fly (e.g. The XX landmark was shifted to the more posterior/anterior/distal/proximal position). The logic explanation which links Fig1 to Fig2 would make the story clear.

5. (line 100, line 193, Fig. 3) The authors seem to measure each area based on the DSRF expression. However, DSRF would be affected by vg knockdown, thereby affecting the measurement. DSRF staining should be used only for detecting wing veins and the outline of wing pouch should be detected by the fold structure surrounding pouch (e.g. using DAPI staining discs). This may resolve the inconsistency of the authors’ data.

6. The authors did not show the clear evidence that an asymmetric cell recruitment actually occurs. G-trace system may be helpful for this issue because G-trace system (vg-Gal4 > G-trace) visualizes current and lineage expression of vg, thereby visualizing the expansion of vg provably induced by autoregulation of vg through cell requirement.

Minor comments

7. The authors should more carefully explain the concept of cell recruitment in wing and the ds-Gal4 > UAS-vg.IR system (eg. The regulation of ds by Vg; short-range induction of vg by adjacent Vg cells, complementary expression of ds and vg,…).

8. (Fig 2B, legend) Are the blue and the orange bars the data values for “control wing vs control-posteriorized/anteriorized wing” and “recruitment-impairment wing vs recruitment-impairment anteriorized/posteriorized wing”, respectively? The authors should describe them carefully as Fig.1.

Reviewer #3: The manuscript by Reyes et al. described the potential role of cell recruitment in the establishment of asymmetry during fly wing development. The authors showed that RNAi knock-down of the selector gene vg resulted in reduction in adult wing asymmetry. Interestingly, the authors found that the reduction of the posterior region was not evident in the larval wing discs. The results are interesting and could partly support the main conclusion. The manuscript is presented in an intelligible fashion and well written. I have a few concerns that the authors should address:

Major:

1. How to explain the discrepancy about the phenotypes observed in the wing disc and the adult wing? If wing discs in later developmental stages, such as late third instar larvae or early pupal stage were dissected, could reduction of the L5-M region be observed?

2. RNAi lines may vary a lot in terms of the knock-down efficiency. Do other vg RNAi lines produce the same phenotype in the adult wing and wing discs?

Minor:

1. In the key words, Drosophila should also be italic.

2. In the title of Figure 3’s legend, “ can not recapitulated” should be “can not be recapitulated”.

Reviewer #4: The manuscript entitled “Cell recruitment and the origin of Anterior-Posterior asymmetry in the Drosophila wing” described the findings that inhibiting vg gene expression during fly wing development led to defects in wing symmetry. The authors argued that cell recruitment mediated by vg contributes to the asymmetric shape of the fly wing. The results are interesting but it is lack of sufficient evidence to support the conclusion. Some major concerns are listed below.

1. The conclusion is based solely on vg RNAi phenotype in the adult wing. The authors should use RNAi or mutants of other key factors/genes to disrupt cell recruitment and examine how the wing asymmetry is affected.

2. The authors need to explain more clearly why different conclusions were made when they and Mun˜oz-Nava et al. used RNAi to knock-down vg and compared the changes of wing shape?

Reviewer #5: Firstly, I congratulate you on your work and results. But some of the details presented in the manuscript are very hard to follow. Some of the details needed to be addressed before the final publication of this manuscript.

1. Was any method employed for testing the validity of genetically crossed flies?

2. Normal D. melanogaster fly could be used as a control group. What is the purpose of using genetically modified flies as a control group?

3. What are the purposes of food-restricted experiments? These details are not much discussed in the results and discussion section of this manuscript.

4. In the case of immunostaining, is it a novel method or a modification of a previously described method? Try to incorporate references if it is a previously described method.

5. In the wing area analysis, sample size is missing. However, in the results section, the authors mentioned the sample size, which is really heterogeneous in nature; a homogenous sample size is required for the comparative analysis.

6. Geometric morphometric analysis and different multivariate statistical analysis could be used to validate the hypothesis which was discussed in the manuscript. However, the author only conducted preliminary shape changes, why?

7. Sample size is very important for shape analysis. Try to include sample size, both right and left wings used for the analysis? Any separate analysis is conducted for the right and left asymmetry analyses?

8. A separate section for statistical analysis can be added to the methodology section.

9. What is the purpose of margin-wise and vein-pairwise comparisons? These details are not much explained in the methodology section.

10. For the comparative analysis, homogenous landmark selection is very important; the significant venation variations also exhibited by genetically modified flies, how the authors arrived at the selection of homogenous landmarks.

11. This manuscript had a separate section for results and discussion. Some of the data are discussed in the results section with proper citations; this kind of writing is normally used in the discussion section, not the results section. The results section only explained the major findings without any comparison with previous findings with references.

12. Based on the previous geometric morphometric analysis, Drosophila wings exhibit a proximal-distal two modular (i.e., modularity) compartmentalization; there is no significant modular organization in AP. Previously, several research articles explained the genetic mechanisms behind wing development and their modular organizations. Try to compare these kinds of findings in the manuscript; it will highlight the novel findings discussed in the manuscript.

Reviewer #6: The study addresses an important question in biology: how the shape of organs is generated and, more specifically, how asymmetries are formed. The authors use the shape and pattern of the Drosophila wing as a model. The authors focus on the antero-posterior (AP) asymmetries, noting that despite symmetric growth signals the wing exhibits a clear AP asymmetry manifested in the position of the veins. The wing primordia cells are specified by Vestigial (Vg), whose expression is controlled by two separate cis-regulatory elements: the vg boundary element (vgBE), which responds to Notch signaling and activates Vg expression at the dorsal-ventral (DV) boundary, and the vg quadrant enhancer (vgQE), which is activated by Wingless (Wg) and Decapentaplegic (Dpp) signaling in the rest of the wing pouch. Importantly, the vgQE is involved in a cell recruitment process that depends on a feed-forward signal sent by Vg-expressing cells to non-expressing cells.

Previous work by the group utilized a genetic system to downregulate cell recruitment by reducing vg expression in hinge cells (ds>vg-RNAi), resulting in smaller, but well-proportioned adult wings. The authors build upon these observations, using quantitative analysis to find that the vg-recruitment-dependent process contributes to generating the asymmetry of adult wing flies, specifically at the L5-M domain. To investigate how cell recruitment in the L5-M region contributes to breaking the symmetry in the wing, they analyzed markers of intervein (DSRF) and vgQE to measure the relative area of each intervein region in larval stages. They conclude that although cell recruitment doesn’t affect the intervein L5-M area in larval discs, this process influences the circularity of the wing pouch and therefore the shape of the adult wing.

The paper is well-written, the logic of the manuscript and their presentation is adequate, and the conclusions are supported by their results. However, I think to properly understand their results and conclusions the authors should introduce better the model of cell recruitment and how they interfere genetically with it.

I have several questions and suggestions that the authors may want to consider before publication.

Major comments:

1. The authors should clearly explain in the introduction, results, and methods sections how cell recruitment to the wing pouch region proceeds during development and how they blocked it. The authors rely on their previous paper (Muñoz-Nava et al., 2020); however, in this new work, very little background and experimental design is described. This part must be improved for the readers to understand this new work.

2. Are there any other factors besides cell recruitment that could contribute to the wing phenotype of ds>vg-RNAi? As Vg has been implicated in cell survival and proliferation, I am not completely convinced that the entire phenotype could be due to cell recruitment. I think this should be discussed and explained.

3. Could the authors provide additional data to convince the reader that the downregulation of Vg is indeed from cells that have been recruited to the Vg domain? Is the ds-Gal4 expressed in cells that have Vg, but not in those that have been actively recruited? As Ds forms a proximo-distal gradient of expression, the authors should clearly demonstrate or at least discuss that in their experiments they are blocking cell recruitment and no other functions of Vg in survival and proliferation.

Minor Comments:

1. In the introduction (line 50), the authors indicate that “a work by Matsuda et al., (2021) has shown that while anterior patterning and growth require Dpp diffusion, it is dispensable in the posterior compartment.” Please correct me if I am wrong, but the work of Matsuda indicates exactly the opposite, that while critical for posterior patterning and growth, Dpp dispersal is largely dispensable for anterior patterning and growth. Please correct.

2. In Fig. 2 legends and result section, explain better how the comparisons were made between the recruitment-impaired wings and the symmetrized versions.

3. The analysis of the wing disc to understand how cell recruitment contributes to breaking the AP asymmetry could be improved. First, could the authors indicate in Fig. 3 which regions of the disc contribute to the adult wing?

4. Line 217: Please correct "invictus" to "invected."

Reviewer #7: This paper investigated the roles of cell recruitment in establishing tissue asymmetry using Drosophila wing development as a model. Wing growth relative to the organ's AP axis was analyzed by using a combination of genetic manipulations and morphometric analyses of wing phenotypes. The paper shows that the impairment of cell recruitment causes a more symmetric adult wing relative to the AP boundary. However, surprisingly, there were local asymmetric patterns of inter-vein growth, which collectively caused the overall symmetric outcomes. While most of the inter-vein regions of the wing showed an increase in growth, the L5-M region of the wing showed a reduction. However, in the larvae, the impairment of cell recruitment did not reduce the L5-M region of the wing imaginal disc from where the adult wing was derived. Instead, the shape of the wing disc pouch was changed. The results suggested some role of cell recruitment in shaping asymmetry, but the exact mechanisms remained elusive. Cell recruitment could be one of the many determinants that redundantly contribute to the overall asymmetric shape of the adult wing.

The general role of cell recruitment in asymmetric organ shapes is interesting. I have a few suggestions:

1) The paper was primarily written for the experts in the field. It would be helpful to provide more descriptions of the various terms and experiments for general readers. For instance, a brief definition/description of cell recruitment in the wing disc context might be more helpful in conceptualizing the experimental results.

2) Lines 142 and 189:

Although a previous paper has been cited on the genetic method used here, a brief description of what genes were manipulated and how is crucial. This would not only enhance the reader's engagement but also provide a more complete picture of the research.

3) What is the relationship of cell recruitment with asymmetric cell division or asymmetric Dpp/Tkv expression/signaling? It would also be helpful to know how the authors ensured that the genetic condition impaired only the cell recruitment process, not the cell division or asymmetric signaling.

4) Discussion (line 233): The discussion could be more clarified.

6. PLOS authors have the option to publish the peer review history of their article (what does this mean?). If published, this will include your full peer review and any attached files.

Reviewer #1: No

Reviewer #2: No

Reviewer #3: No

Reviewer #4: No

Reviewer #5: No

Reviewer #6: No

Reviewer #7: No

---

## [Author Response · Author response to Decision Letter 0]

17 Sep 2024

Please view the Response to Reviewers file included in this submission.

---

## [Decision Letter · Decision Letter 1]

18 Oct 2024

Cell recruitment and the origins of Anterior-Posterior asymmetries in the Drosophila wing

PONE-D-24-23042R1

Dear Dr. Nahmad,

We’re pleased to inform you that your manuscript has been judged scientifically suitable for publication and will be formally accepted for publication once it meets all outstanding technical requirements.

Kind regards,

Abhinava Kumar Mishra, PhD

Academic Editor

PLOS ONE

Additional Editor Comments (optional):

Reviewers' comments:

Reviewer's Responses to Questions

**Comments to the Author**

1. If the authors have adequately addressed your comments raised in a previous round of review and you feel that this manuscript is now acceptable for publication, you may indicate that here to bypass the “Comments to the Author” section, enter your conflict of interest statement in the “Confidential to Editor” section, and submit your "Accept" recommendation.

Reviewer #2: All comments have been addressed

Reviewer #3: All comments have been addressed

Reviewer #5: All comments have been addressed

Reviewer #6: All comments have been addressed

2. Is the manuscript technically sound, and do the data support the conclusions?

Reviewer #2: Yes

Reviewer #3: Yes

Reviewer #5: Yes

Reviewer #6: Yes

3. Has the statistical analysis been performed appropriately and rigorously? 

Reviewer #2: Yes

Reviewer #3: Yes

Reviewer #5: Yes

Reviewer #6: Yes

4. Have the authors made all data underlying the findings in their manuscript fully available?

Reviewer #2: Yes

Reviewer #3: Yes

Reviewer #5: Yes

Reviewer #6: Yes

5. Is the manuscript presented in an intelligible fashion and written in standard English?

Reviewer #2: Yes

Reviewer #3: Yes

Reviewer #5: Yes

Reviewer #6: Yes

6. Review Comments to the Author

Reviewer #2: The authors have addressed my concerns adequately, and the revised manuscript has been improved well. Although the authors have not clearly explained the discrepancy of the phenotype between larval and adult wings yet, I considered that additional experiments are not entirely needed in this paper.

Reviewer #3: The authors have addressed my concerns in the revised manuscript, I have no further questions. But I suggest that the authors should avoid over-stating the role of vg in "origins of Anterior-Posterior asymmetries".

Reviewer #5: (No Response)

Reviewer #6: I congratulate the authors for the effort made in responding to the questions and comments from each of the seven reviewers.

7. PLOS authors have the option to publish the peer review history of their article (what does this mean?). If published, this will include your full peer review and any attached files.

Reviewer #2: No

Reviewer #3: No

Reviewer #5: No

Reviewer #6: **Yes: **Carlos Estella

---

## [Editor Report · Acceptance letter]

28 Oct 2024

PONE-D-24-23042R1 

PLOS ONE

Dear Dr. Nahmad, 

I'm pleased to inform you that your manuscript has been deemed suitable for publication in PLOS ONE. Congratulations! Your manuscript is now being handed over to our production team.

Kind regards, 

on behalf of

Dr. Abhinava Kumar Mishra 

Academic Editor

PLOS ONE